# Validation and Establishment of the SARS-CoV-2 Lentivirus Surrogate Neutralization Assay as a Prescreening Tool for the Plaque Reduction Neutralization Test

John Merluza,[a] Johnny Ung,[a] Kai Makowski,[a] Alyssia Robinson,[a] Kathy Manguiat,[a] Nicole Mueller,[a] Jonathan Audet,[a] Julie Chih-Yu Chen,[a,b] James E. Strong,[a,c,d] Heidi Wood,[a] Alexander Bello[a]

aPublic Health Agency of Canada, National Microbiology Laboratory, Winnipeg, Manitoba, Canada
bDepartment of Biochemistry and Medical Genetics, University of Manitoba, Winnipeg, Manitoba, Canada
cDepartment of Pediatrics & Child Health, University of Manitoba, Winnipeg, Manitoba, Canada
dDepartment of Medical Microbiology & Infectious Diseases, University of Manitoba, Winnipeg, Manitoba, Canada

**ABSTRACT** Neutralization assays are important for understanding and quantifying neutralizing antibody responses toward severe acute respiratory syndrome coronavirus 2 (SARS-CoV-2). The SARS-CoV-2 lentivirus surrogate neutralization assay (SCLSNA) can be used in biosafety level 2 (BSL-2) laboratories and has been shown to be a reliable alternative approach to the plaque reduction neutralization test (PRNT). In this study, we optimized and validated the SCLSNA to assess its ability as a comparator and pre-screening method to support the PRNT. Comparability between the PRNT and SCLSNA was determined through clinical sensitivity and specificity evaluations. Clinical sensitivity and specificity assays produced acceptable results, with 100% (95% confidence interval [CI], 94% to 100%) specificity and 100% (95% CI, 94% to 100%) sensitivity against ancestral Wuhan spike-pseudotyped lentivirus. The sensitivity and specificity against B.1.1.7 spike-pseudotyped lentivirus were 88.3% (95% CI, 77.8% to 94.2%) and 100% (95% CI, 94% to 100%), respectively. Assay precision measuring intra-assay variability produced acceptable results for high (50% PRNT [$PRNT_{50}$], 1:≥640), mid ($PRNT_{50}$, 1:160), and low ($PRNT_{50}$, 1:40) antibody titer concentration ranges based on the $PRNT_{50}$, with coefficients of variation (CVs) of 14.21%, 12.47%, and 13.28%, respectively. Intermediate precision indicated acceptable ranges for the high and mid concentrations, with CVs of 15.52% and 16.09%, respectively. However, the low concentration did not meet the acceptance criteria, with a CV of 26.42%. Acceptable ranges were found in the robustness evaluation for both intra-assay and interassay variability. In summary, the validation parameters tested met the acceptance criteria, making the SCLSNA method fit for its intended purpose, which can be used to support the PRNT.

**IMPORTANCE** Neutralization studies play an important role in providing guidance and justification for vaccine administration and helping prevent the spread of diseases. The neutralization data generated in our laboratory have been included in the decision-making process of the National Advisory Committee on Immunization (NACI) in Canada. During the coronavirus 2019 (COVID-19) pandemic, the plaque reduction neutralization test (PRNT) has been the gold standard for determining neutralization of SARS-CoV-2. We validated a SARS-CoV-2 lentivirus surrogate neutralization assay (SCLSNA) as an alternative method to help support the PRNT. The advantages of using the SCLSNA is that it can process more samples, is less tedious to perform, and can be used in laboratories with a lower biosafety level. The use of the SCLSNA can further expand our capabilities to help fulfill the requirements for NACI and other important collaborations.

Address correspondence to Alexander Bello, alexander.bello@phac-aspc.gc.ca.

The authors declare no conflict of interest.

**KEYWORDS** plaque reduction neutralization test, SARS-CoV-2 lentivirus surrogate neutralization assay, validation

The coronavirus 2019 (COVID-19) pandemic has caused 448,624,192 confirmed cases and 6,507,879 deaths worldwide as of 9 September 2022, an unprecedented number (https://www.worldometers.info/coronavirus/). However, the rapid development and administration of vaccines such as Pfizer-BioNTech and Moderna have contributed to helping prevent severe disease and mortality among infected individuals (1–3). As the COVID-19 pandemic unfolded over time, it was shown that the spike glycoprotein found in the severe acute respiratory syndrome coronavirus 2 (SARS-CoV-2) virus membrane can undergo mutations, resulting in variants that can evade neutralizing antibodies (NAbs) generated against previous iterations of the spike protein, leading to new waves of infection (4, 5). Breakthrough infections have been a challenge throughout the pandemic, and neutralization studies are important for analyzing the neutralizing antibody response, which plays an essential role in preventing severe infection, and for assessing vaccine candidate suitability (6, 7).

The plaque reduction neutralization test (PRNT) is the current gold-standard neutralization assay; however, this method is labor-intensive and requires the use of a biosafety level 3 (BSL-3) or higher containment laboratory (8–12). In addition, the PRNT assay relies on visualization of plaques formed by the virus, resulting in a longer turnaround time (TAT) from sample receipt to result (13, 14). Such limitations present challenges to sample processing and throughput capabilities, and alternate methodologies are required to help circumvent these difficulties. The SARS-CoV-2 lentivirus surrogate neutralization assay (SCLSNA) is one such approach that does not have the same logistical challenges as the PRNT assay. SCLSNA can be safely performed in BSL-2 laboratories; it is amenable to high throughput and has a relatively faster TAT of 48 h (15–18). The SCLSNA incorporates the use of lentiviruses pseudotyped with SARS-CoV-2 spike protein, which can serve as a surrogate virus to quantitate neutralizing antibodies generated against the SARS-CoV-2 spike protein (19, 20). The lentivirus particles used in this study are second-generation lentiviral vectors that do not contain accessory virulence genes such as *vif*, *vpu*, and *nef*, rendering them replication incompetent and allowing for their safe use in a BSL-2 laboratory (21).

In this study, we performed a method validation to determine if the SCLSNA was fit for its intended purpose as a reliable comparator and screening method to complement the PRNT (22). Following guidelines recommended by the WHO and the Food and Drug Administration (FDA), this study targeted validation parameters such as precision, repeatability, robustness, linearity, limit of detection (LOD), and limit of quantification (LOQ) (22, 23). We optimized the SCLSNA to confirm the optimal assay parameter conditions and to limit variation, as well as to assess the clinical sensitivity and specificity in comparison to the PRNT.

## RESULTS

**Cell-seeding optimization.** A cell-seeding optimization experiment for HEK293T/ACE2-TMPRSS2 cells was performed to determine the optimal sensitivity for the SCLSNA, while trying to maximize pseudotyped lentivirus infection and minimize the variation between sample replicates. A high-titer sample (50% plaque reduction/neutralization titer [$PRNT_{50}$], 1:≥640) was tested against an ancestral Wuhan spike-pseudotyped lentivirus with nine cell-seeding densities, ranging from $7.8 \times 10^2$ to $2.0 \times 10^5$ cells/well (Fig. 1A). The selection for the optimal cell density was based on a combination of the cell density (>$1 \times 10^3$ cells/well), half-maximal inhibitory concentration ($IC_{50}$; >640), and goodness of fit ($R^2 > 0.9$). The results indicate $IC_{50}$ values of >640 and $R^2$ values of >0.9 for cell densities between $7.8 \times 10^2$ and $6.3 \times 10^3$, but these seeding densities were not selected due to the potential for increased variability in SCLSNA testing observed with the lower cell densities (9). Cell densities above $1 \times 10^4$ cells/well demonstrated reduced $IC_{50}$ or $R^2$

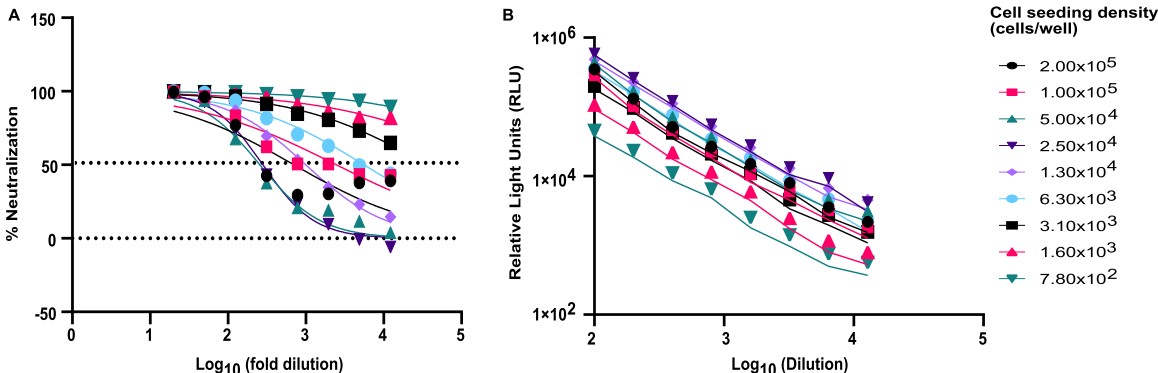

**FIG 1** (A) Cell density optimization for neutralization. Cell density experiments were performed to determine the optimal cell numbers based on neutralization. Nine different cell concentrations were used and performed on three plates. The average RLU values were determined for each cell concentration and used to calculate the IC$_{50}$ and $R^2$ using GraphPad Prism v.9.3 software. (B) Pseudovirus titration against different cell-seeding densities to determine the optimal pseudovirus signal in relation to the cell-seeding density. Ancestral Wuhan spike-pseudotyped lentivirus was diluted 100-fold, followed by an 8-step 2-fold serial dilution. RLU values were used to determine the optimal cell-seeding density. HEK293T/ACE2-TMPRSS2 cells and lentivirus were incubated in a 5% CO$_2$ incubator at 37°C for 18 to 24 h prior to the detection of RLU.

values. Thus, a cell density of $1.0 \times 10^4$ cells/well was selected as the optimal cell-seeding density for the SCLSNA (Table 1).

**Pseudovirus titration.** Pseudovirus titration using ancestral Wuhan spike-pseudotyped lentivirus was performed to identify the optimal cell density corresponding to a high pseudovirus relative luminescence units (RLU) signal. Establishing a high pseudovirus RLU signal is required to create a sufficient signal above the cell-only background of at least 1,000-fold, in order to determine reportable IC$_{50}$ values that meet the acceptance criteria (19). The pseudovirus was initially diluted 100-fold, followed by an 8-step 2-fold serial dilution. A decrease in the pseudovirus RLU signal was shown at cell densities above $2.5 \times 10^4$ cells/well and below $1.3 \times 10^4$ cells/well. High RLU values were observed at $1.30 \times 10^4$ cells/well, showing a linear response from the serial dilutions, which were used to justify the cell density selection of $1.30 \times 10^4$ cells/well (Fig. 1).

**Clinical specificity and sensitivity.** The specificity and sensitivity were examined against the ancestral Wuhan and B.1.1.7 spike-pseudotyped lentiviruses, in comparison to the gold-standard PRNT assay. Sixty samples positive for SARS-CoV-2 from the National Microbiology Laboratory (NML) COVID-19 National Panel and 60 SARS-CoV-2-negative pre-COVID-19 samples were used for the experiment. The results for both parameters against ancestral Wuhan spike-pseudotyped lentivirus were acceptable, achieving 100% (95% CI, 94% to 100%) specificity and 100% (95% CI, 94% to 100%) sensitivity. For B.1.1.7 spike-pseudotyped lentivirus, a sensitivity of 88.3% (95% CI, 77.8% to 94.2%) and specificity of 100% (95% CI, 94% to 100%) were achieved (Table 2). Perfect interrater agreement with the PRNT$_{50}$ was demonstrated against the ancestral Wuhan spike-pseudotyped lentivirus, and almost perfect

**TABLE 1** Optimal IC$_{50}$ determination based on cell-seeding density and $R^2$ value[a]

| Cell density (cells/well) | IC$_{50}$ | $R^2$ |
|---|---|---|
| $2.00 \times 10^5$ | 703.4 | 0.7144 |
| $1.00 \times 10^5$ | 2472 | 0.8515 |
| $5.00 \times 10^4$ | 256.8 | 0.9705 |
| $2.50 \times 10^4$ | 280.7 | 0.9929 |
| $1.30 \times 10^4$ | 997.8 | 0.9917 |
| $6.30 \times 10^3$ | $5.58 \times 10^3$ | 0.9658 |
| $3.10 \times 10^3$ | $3.70 \times 10^4$ | 0.9782 |
| $1.60 \times 10^3$ | $3.05 \times 10^5$ | 0.9083 |
| $7.80 \times 10^2$ | $1.00 \times 10^6$ | 0.9795 |

[a]GraphPad Prism v.9.3 was used to determine the IC$_{50}$ and goodness-of-fit ($R^2$) values.

**TABLE 2** Results of the comparison of clinical specificity and sensitivity between the SCLSNA and the PRNT$_{50}$

| Category | Lentivirus | No. (%) of samples | | | PPV[a] | NPV | Accuracy (%) | Precision (%) | Cohen's kappa |
| --- | --- | --- | --- | --- | --- | --- | --- | --- | --- |
| | | Total | Positive | Negative | | | | | |
| Positive SARS-CoV-2 patients | Ancestral | 60 | 60 (100) | 0 (0) | 100 | 100 | 100 | 100 | 0.100 |
| Prepandemic adult patients | Ancestral | 60 | 0 (0) | 60 (100) | | | | | |
| Positive SARS-CoV-2 patients | B.1.1.7 | 60 | 53 (88.3) | 7 (11.7) | 88 | 100 | 94 | 88 | 0.883 |
| Prepandemic adult patients | B.1.1.7 | 60 | 0 (0) | 60 (100) | | | | | |

[a]PPV, positive predictive value; NPV, negative predictive value.

agreement ($\kappa$, 0.883) was shown with the B.1.1.7 spike-pseudotyped lentivirus against the PRNT$_{50}$ (Table 2).

**Validation of the SCLSNA.** Guidelines used for the validation were based on those by the WHO (23) and FDA (24). The validation parameters assessed in this study included precision (repeatability, intermediate precision), robustness, linearity, limit of detection, and limit of quantification, as described below. Accuracy was not assessed due to the limitation in accurately comparing reportable values between the IC$_{50}$ values of the SCLSNA and PRNT$_{50}$.

**(i) Precision.** *(a) Repeatability (intra-assay precision).* Repeatability was measured using three concentrations that were based on our in-house PRNT$_{50}$ titer results. The concentrations consisted of high-titer (PRNT$_{50}$, 1:≥640), mid-titer (PRNT$_{50}$, 1:160), and low-titer (PRNT$_{50}$, 1:40) samples. Analysts processed each sample in triplicate during three separate weeks for a total of nine determinations each (Fig. 2). The coefficient of variation (CV) percentage for each concentration was within the acceptance criteria of ≤20%, with values of 14.21% (high titer), 12.47% (mid titer), and 13.28% (low titer). Weekly comparisons were within the acceptable range, with CV percentages of 14.44% (week 1), 18.79% (week 2), and 9.696% (week 3).

*(b) Intermediate precision (interassay variability).* Interassay variability was assessed by comparing the same homogeneous sample between different analysts tested during different weeks. Each analyst tested six replicates of high-, mid-, and low-titer samples from the NML CNP in three separate weeks for 18 determinations (Fig. 2, 3). The percent CV for high- and mid-titer samples between analysts were within the acceptable range, with CVs of 15.52% (high titer) and 16.09% (mid titer). The percent CV for low-titer samples did not meet the acceptance criteria, with a CV of 26.42%, indicating slightly higher variation within the low-titer samples between analysts.

**(ii) Robustness.** The robustness of the SCLSNA was examined to measure the ability of the procedure to provide analytical results of acceptable accuracy and precision under a variety of conditions. In this experiment, high-, mid-, and low-titer samples were tested in triplicate for two independent runs, and the detection method was compared using an Agilent BioTek Cytation 1 cell-imaging multimode reader device and Promega's GloMax Navigator microplate luminometer (Fig. 2). Results from the different devices were compared to determine if they would be influenced by changes in the operational conditions. The intra-assay variability for each device was below 20% CV and acceptable (Table 3). The interassay variability between the devices for each concentration was below 20% CV and passed the criteria (Table 3).

**(iii) Linearity.** Linearity was assessed using a WHO international reference panel for anti-SARS-CoV-2 immunoglobulin. The WHO panel consisted of five pooled human plasma samples of high-titer (20/150), mid-titer (20/148), low-titer 1 (20/144), low-titer 2 (20/140), and pre-COVID-19 (20/142) samples. IC$_{50}$ titers obtained from the SCLSNA indicated that they were directly proportional to the antibody titers of the WHO reference panel, with IC$_{50}$ values of 331.2 (high), 171.7 (mid), 107.8 (low 1), 32.01 (low 2), and 10 (pre-COVID-19) (Fig. 4A and B). SCLSNA IC$_{50}$ values and antibody titers (IU/mL) from the WHO reference panel were compared using Pearson's correlation coefficient analysis. A strong correlation between the WHO reference panel antibody titers and SCLSNA was observed, with a correlation of $r = 0.9210$ ($P = 0.0263$) (Fig. 4C). Linearity

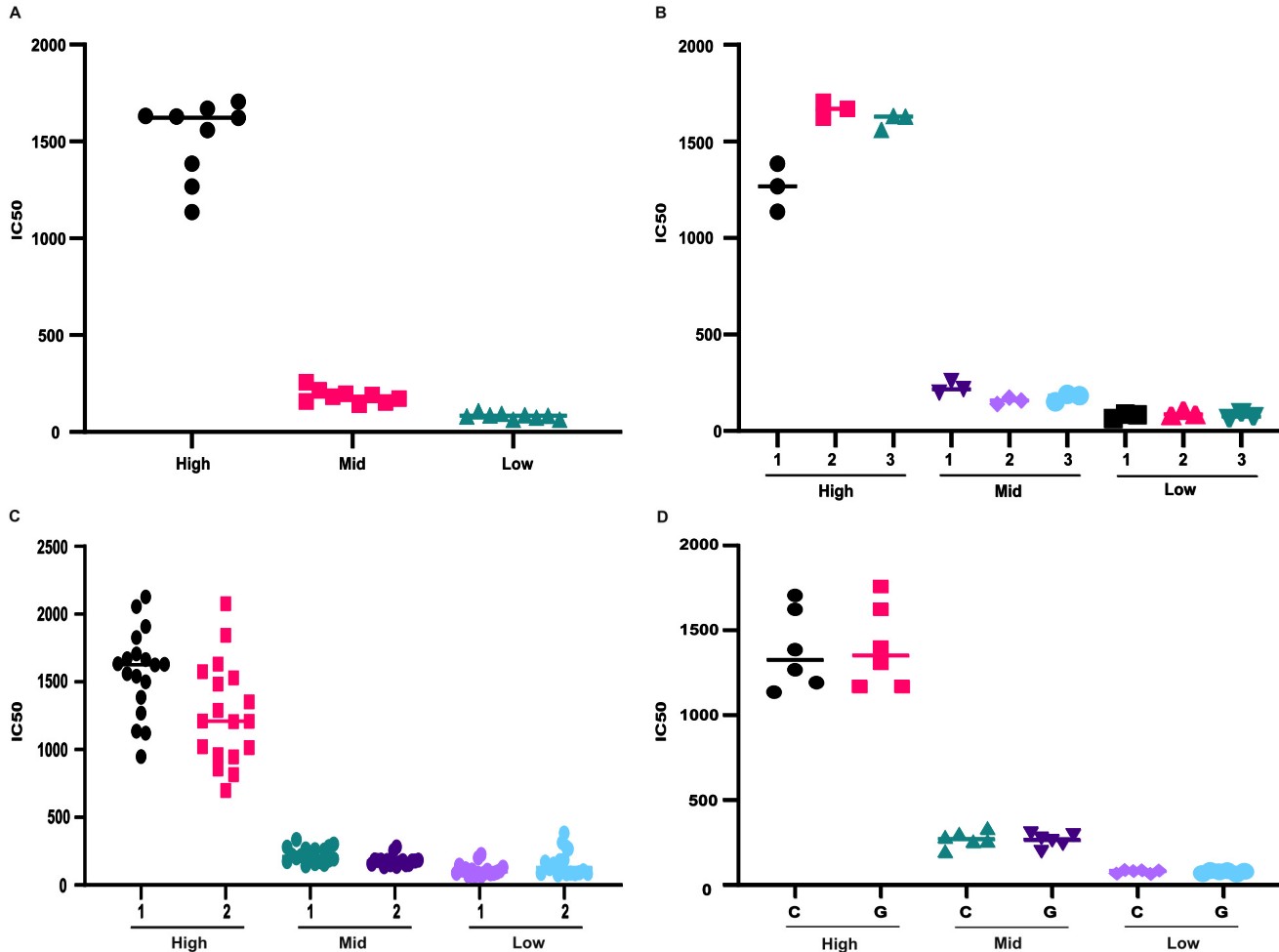

**FIG 2** (A, B) Intra-assay variability of SCLSNA. One analyst tested high-titer (PRNT$_{50}$, 1:≥640), mid-titer (PRNT$_{50}$, 1:160), and low-titer (PRNT$_{50}$, 1:40) samples from the NML CNP against ancestral Wuhan spike-pseudotyped lentivirus. (A) Analyst 1 tested samples in triplicate during three separate weeks for nine determinations of each concentration. IC$_{50}$ values were compared and evaluated based on the percent CV. (B) Analyst 1 week-to-week comparison, measuring intra-assay variability using the same samples along with the same conditions and equipment each week. (C) Interassay variability of SCLSNA between two analysts. Two analysts tested high-, mid-, and low-titer samples from the NML CNP against ancestral Wuhan spike-pseudotyped lentivirus. Samples were tested in six replicates during three separate weeks for a total of 18 determinations each. The solid line represents the mean IC$_{50}$ titer. Results were reported as IC$_{50}$ titers, and percent CV comparisons between analysts were conducted using GraphPad Prism v.9.3 software. (D) Interassay variability comparison of the IC$_{50}$ value between the Agilent BioTek Cytation 1 and Promega's GloMax Navigator microplate luminometer (labeled C and G, respectively). Analyst 1 tested samples with six replicates for each device, totaling 18 determinations. The solid line represents the mean IC$_{50}$ titer. Results were reported as IC$_{50}$ titers, and percent CV comparisons between devices were conducted using GraphPad Prism v.9.3 software.

was also assessed with pseudovirus addition and RLU. Here, we showed dilutional linearity with the pseudovirus that was observed with decreasing RLU as the dilution increased (Fig. 4D).

**(iv) Limits of detection and quantification.** The LOD and LOQ for the SCLSNA were determined using 36 samples from the NML CNP that were collected pre-COVID-19 and negative for SARS-CoV-2 (as verified by the PRNT). The standard deviation determined from the mean IC$_{50}$ values of the negative samples resulted in an LOD of 19.60 and an LOQ of 65.32, which were 3 and 10 times the standard deviation, respectively. We used a cutoff of <20 for negative samples and assigned them a nominal value of 10. This was done to distinguish negative from positive results in our qualitative representation of our results.

## DISCUSSION

We have shown the SCLSNA to be a suitable alternative to the gold-standard PRNT. In this validation study, we established acceptable validation parameters for precision,

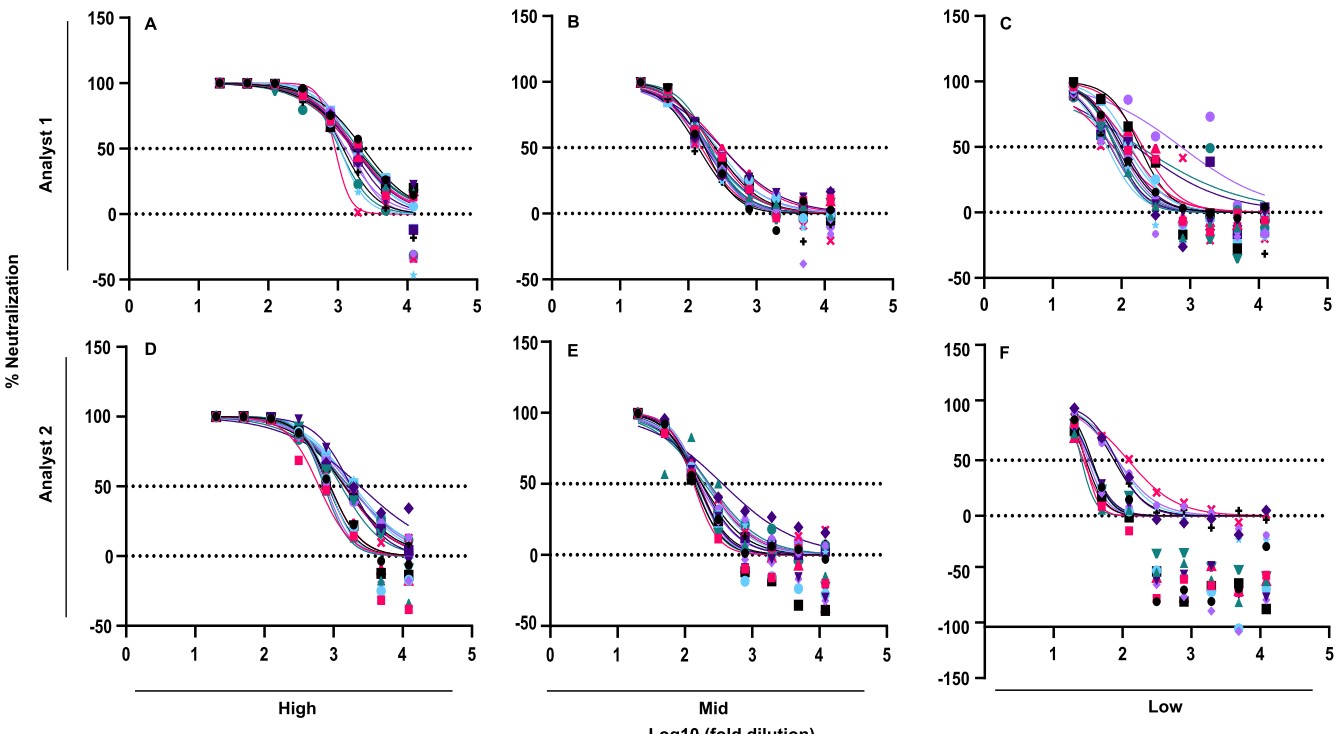

**FIG 3** Interassay variability of SCLSNA between two analysts. Percent neutralization comparison between two analysts using high-titer (PRNT$_{50}$, 1:≥640) (A, D), mid-titer (PRNT$_{50}$, 1:160) (B, E), and low-titer (PRNT$_{50}$, 1:40) (C, F) samples from the NML CNP against ancestral Wuhan spike-pseudotyped lentivirus. Samples were tested in six replicates during three separate weeks for a total of 54 determinations. IC$_{50}$ titers were determined using GraphPad Prism v.9.3 software.

robustness, and linearity, while optimizing and displaying sensitivity and specificity with the SCLSNA that were comparable to the PRNT (Fig. 5). Other studies have previously validated similar versions of surrogate neutralization assays, but the goal of this study was to expand the validation parameters tested and include a sample concentration range based off the PRNT to further confirm the reliability and strength of the SCLSNA as a comparable approach to the PRNT (9, 10, 17).

Overall, good precision was shown throughout the validation study. Previous studies have also shown good precision for both intra-assay and interassay variability (9, 10, 25, 26), but one key difference in our approach was the use of a broad concentration range of samples. The incorporation of high-titer (PRNT$_{50}$, 1:≥640), mid-titer (PRNT$_{50}$, 1:160), and low-titer (PRNT$_{50}$, 1:40) samples allowed us to directly compare samples between the SCLSNA and PRNT, allowing for a thorough analysis of precision within and between analysts. Neerukonda et al. used a similar broad-based approach to sample concentrations; however, more variation was detected in their intermediate precision, greater than that shown in our study (17). We also detected higher than expected variation among the low-titer samples between analysts, which may be due to the specificity of binding inherent within the SCLSNA, which focuses solely on the receptor binding

**TABLE 3** Intra-assay variability analysis and interassay variability using the Agilent BioTek Cytation 1 and Promega's GloMax Navigator microplate luminometer with high-, mid-, and low-titer samples[a]

| Characteristic | Device | Data by titer: | | |
| --- | --- | --- | --- | --- |
| | | High | Mid | Low |
| %CV | Cytation1 | 16.87 | 16.84 | 12.77 |
| %CV | GloMax Navigator | 17.27 | 15.18 | 11.61 |
| Interassay variability | | 0.9634 | 3.027 | 1.441 |

[a]Samples were tested in six replicates on each device, for 18 determinations each.

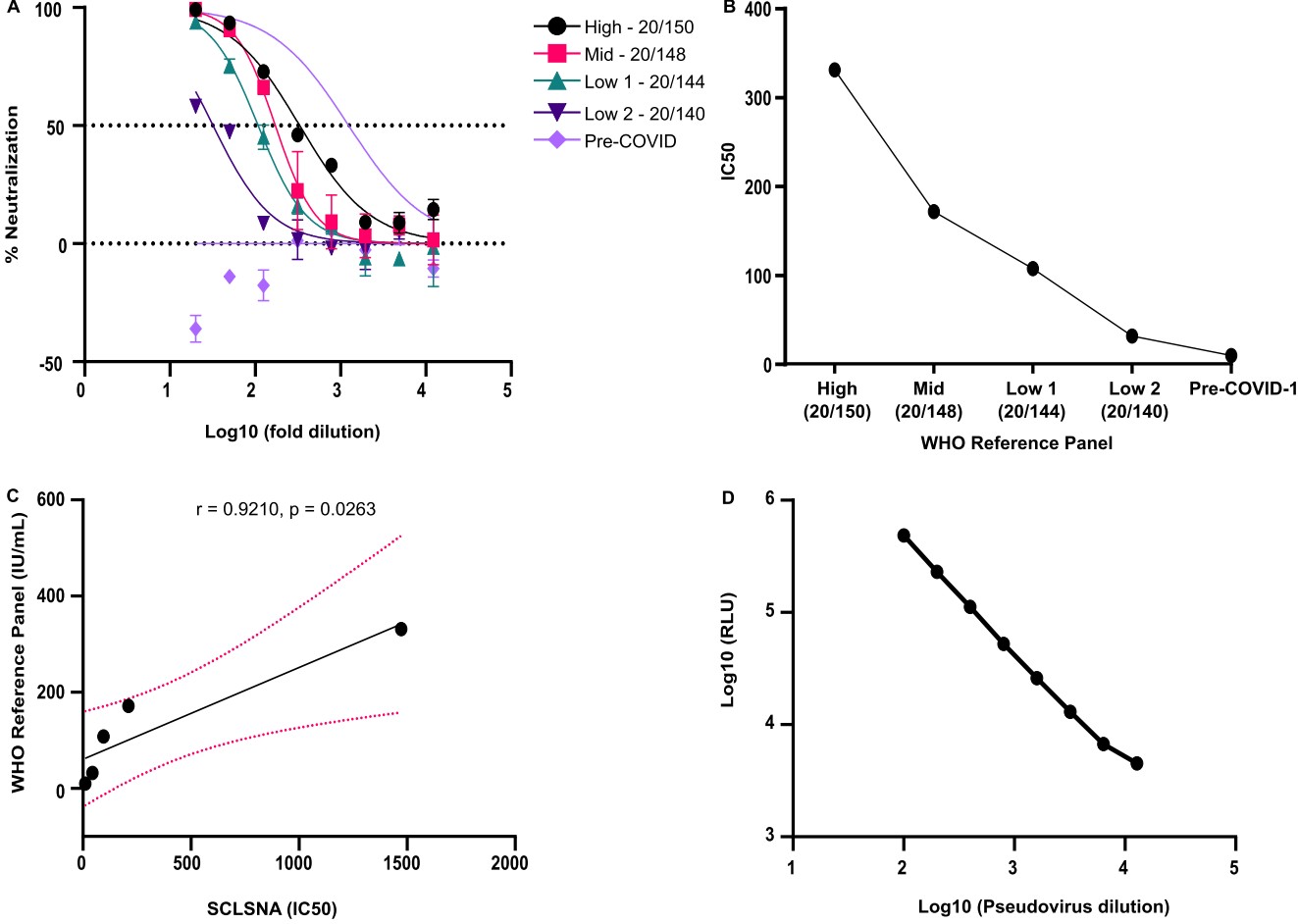

**FIG 4** Linearity analysis of the SCLSNA. (A) SCLSNA analysis of high-, mid-, and low-titer and pre-COVID-19 samples. $IC_{50}$ titers were determined using GraphPad Prism v.9.3. (B) A linearity analysis was performed to compare the $IC_{50}$ from the SCLSNA to the antibody titers from the WHO reference panel. Five reference standards were tested (high titer, mid-titer, low titer 1, low titer 2, and pre-COVID-19). (C) Correlation analysis between the SCLSNA and the WHO reference panel (IU/mL). The solid line indicates the regression line, and the dashed line indicates the 95% confidence interval (CI) (0.2066 to 0.9949). Pearson's correlation coefficient and $P$ value are indicated. (D) Linearity analysis of pseudovirus dilution and relative luminescence units.

domain and spike regions of SARS-CoV-2, as opposed to nonspecific binding to live SARS-CoV-2 cells that may be found in plasma samples at lower dilutions (27, 28). Spike protein density within a pseudotyped lentivirus may also be different than in live SARS-CoV-2 cells, which may result in a decreased amount of neutralization, particularly in low-titer samples (29). Despite the outcome observed in the low-titer sample comparison, we were still able to show acceptable precision for the intra- and interassay variability in each concentration range, along with low variation within and between analysts.

Another strategy we employed in our validation was the use of two different luminescence detectors to achieve robustness; we were able to achieve acceptable robustness from the low variation seen across different luminescence detectors. To our knowledge, this approach has not been taken in previous studies; we showed low variation across several different devices, indicating the flexibility of the SCLSNA in its performance and capabilities. One limitation of our robustness analysis was the inability to compare assay performance in different laboratories, due to logistical challenges and unavailability at the time of the study. We opted to test robustness using different luminescence detectors as an alternative and to confirm the reliability of the assay.

The linearity of the SCLSNA was demonstrated after comparison with the WHO international reference panel. The SCLSNA closely approximated the expected concentrations of the standard, showing a strong correlation between the SCLSNA and NIBSC

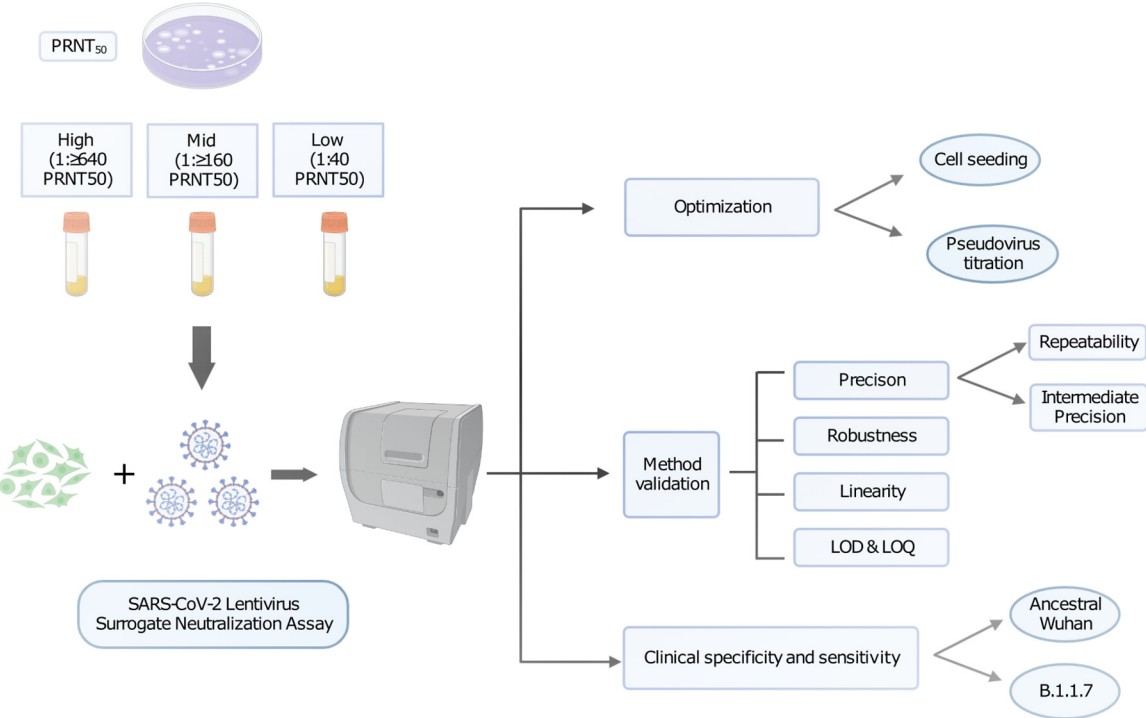

**FIG 5** Flow chart schematic of the SCLSNA validation study design. High-titer (PRNT$_{50}$, 1:≥640), mid-titer (PRNT$_{50}$, 1:160), and low-titer (PRNT$_{50}$, 1:40) samples from the NML CNP were tested against ancestral Wuhan spike-pseudotyped lentivirus by the SCLSNA. Assay optimization was conducted to determine the optimal HEK293T/ACE2-TMPRSS2 cell-seeding density, and a pseudovirus titration was performed to confirm the optimal cell-seeding density. All high-, mid-, and low-titer samples were used for the method validation to test the precision, robustness, linearity, LOD, and LOQ. Direct comparison of the SCLSNA to the gold-standard PRNT was made using the clinical specificity and sensitivity. Created with BioRender.com.

reference standard. A study conducted by Yu et al. achieved dilutional linearity, which was also confirmed in our study, highlighting the ability of the SCLSNA to measure expected values that are directly proportional to the amount of pseudovirus used (26).

Optimization of the SCLSNA established the optimal seeding density, as previously shown in other studies (9, 10). In those studies, the Huh7 and BHK21-hACE2 cell lines were used, in contrast to our studies, where we opted to use HEK293T/ACE2-TMPRSS2 cells due to the ability of TMPRSS2 to prime spike proteins on pseudotyped lentiviruses, likely increasing infectivity (30). Another important component of this analysis was the addition of Polybrene, which is a polycationic agent that helps facilitate pseudovirus cell entry (31). In our study, we optimized the SCLSNA to help establish consistency, minimize variation, and ensure that our method was performing at optimal levels.

To determine if the SCLSNA could perform similarly to the gold-standard PRNT, we conducted a study comparing the SCLSNA to the PRNT through analysis of the clinical sensitivity and specificity. Using either ancestral Wuhan or B.1.1.7 spike-pseudotyped lentivirus, we were able to achieve acceptable sensitivity and specificity. We observed very high sensitivity and specificity for ancestral Wuhan spike-pseudotyped lentivirus; however, we also saw a decrease in sensitivity with the B.1.1.7 spike-pseudotyped lentivirus. This may be attributed to a reduced level of neutralization found against the B.1.1.7 variant, as evidenced by reduced neutralization activities of various monoclonal antibodies (32, 33). In addition, convalescent-phase sera and vaccine-induced antibody responses are still effective against the B.1.1.7 variant, but the immune response may vary in comparison to the ancestral Wuhan pseudotyped lentivirus (32, 33). To determine the correlation between the SCLSNA and PRNT using samples from convalescent SARS-CoV-2 patients, we conducted a correlation assessment, but the correlation coefficient was low (data not shown). The NML COVID-19 National Panel sample set used in this

study consisted mainly of one antibody titer range at lower neutralization titers (PRNT$_{50}$, 1:80) from convalescent donors who were naturally infected with SARS-CoV-2 before the vaccine was available, making it difficult to achieve a proper correlation analysis. As a result, we used clinical sensitivity and specificity as a measure for comparison and were able to show good comparability to the PRNT, as seen in other studies (12, 25).

This validation study successfully achieved acceptable criteria in all the parameters tested, proving the SCLSNA to be a reliable prescreening approach to the PRNT. One of the key advantages to the SCLSNA is the use of a SARS-CoV-2 pseudotyped lentivirus generation platform. The lentivirus system enables an efficient and quick TAT for generating lentiviruses pseudotyped with the target of interest. This is particularly important during the ongoing pandemic, with the emergence of novel variants of concern, against which preexisting NAbs may be less effective. As well, all plasmids used for generating our pseudotyped lentiviruses are commercially available, making for a convenient and time-saving approach in comparison to custom-designed plasmids, which are more time-consuming to prepare (15). In addition, the in-house generation of pseudotyped lentiviruses is a faster approach than the generation of live virus in a BSL-3 setting, because it may take time to successfully rescue live virus and optimize the assay conditions.

Another key advantage is the quantitative output of the SCLSNA. The data generated from this assay give a more precise antibody titer with the IC$_{50}$, rather than the visual determination of antibody titer provided by the PRNT method, which is considered more subjective, as technical staff must be carefully trained to accurately identify plaque formation by visual means (25). It is also difficult to obtain an endpoint dilution for a large sample set, especially ones containing high antibody titers, resulting in a broad estimation of antibody titer and a cutoff value assigned in the reported data affecting the overall precision of the results. Data generated by the PRNT can be more subjective across different analysts, further decreasing the accuracy and consistency of results (8). Furthermore, sample throughput for the PRNT is limited to processing a smaller number of samples, because it requires plates with larger well sizes and manual labor (16, 34). The plaque morphology changes with each new variant, resulting in subjectivity among analysts and inaccurate reporting of results (35, 36). The focus reduction neutralization test (FRNT) has been used as an alternative to the PRNT, but limitations such as the need for a BSL-3 facility and qualitative analysis still remain (37–39). In contrast, the SCLSNA can be used for high-throughput, automated sample processing in 96- to 384-well plate formats (16, 34).

Validation of the SCLSNA provides an alternative neutralizing antibody platform to support or potentially replace the PRNT gold-standard method. The SCLSNA does not require the handling of live SARS-CoV-2 virus in a BSL-3 facility, providing for a safer work environment, is less tedious, and has a faster TAT for sample processing to reporting of results. The quantitative analysis that is achievable by the SCLSNA increases its precision, making it a reliable approach to the limitations found inherent within the PRNT. The validation parameters tested in this study met the previously established acceptance criteria, making the SCLSNA a suitable alternative to the PRNT.

## MATERIALS AND METHODS

**Study population and specimen collection.** Plasma samples used in the validation were obtained from the National Microbiology Laboratory COVID-19 National Panel (NML CNP) under the approval of the Research Ethics Board (REB-2020-004P). Samples from patients who had previously tested positive for SARS-CoV-2 by reverse transcription-quantitative PCR (RT-qPCR) were included in the NML COVID-19 National Panel. Blood sample collection took place from 13 May 2020 to 22 August 2020. All samples were collected from various provinces nationwide through the Canadian Blood Services (40). Plasma samples were heat-inactivated for 30 min at 56°C and then stored at −80°C until testing was performed.

**Cell lines.** For the SCLSNA, HEK293T/ACE2-TMPRSS2 cells (GeneCopoeia, Rockville, MD) were used for infection by pseudotyped lentivirus. These cells stably express angiotensin-converting enzyme 2 (ACE2) and transmembrane serine protease 2 (TMPRSS2), which are important for infection by SARS-CoV-2 and other pseudotyped viruses expressing SARS-CoV-2 spike on their surface.

For pseudotyped lentivirus production, AAVpro 293T cells (TaKaRa Bio, San Jose, CA) were used to transfect the envelope plasmid, transfer plasmid, and packaging plasmid. All cell lines were incubated in

a 5% $CO_2$ incubator at 37°C with Dulbecco's modified Eagle medium (DMEM; Gibco, Waltham, MA) supplemented with 10% heat-inactivated fetal bovine serum (FBS), 1% penicillin/streptomycin, 1% L-glutamine, and 1% sodium pyruvate (DMEM10) (Gibco).

**SARS-CoV-2 pseudotyped virus production.** All assays and lentivirus preparations were performed in BSL-2 conditions unless noted differently. AAVpro 293T cells (TaKaRa Bio) were used to generate the SARS-CoV-2 spike pseudotyped lentiviruses in 10- to 150-mm Corning dishes (Millipore Sigma, St. Louis, MO). Briefly, psPAX2 empty-vector HIV packaging plasmid (Addgene, Watertown, MA; a gift from Didier Trono), SARS-CoV-2 spike protein (ancestral Wuhan or B.1.1.7) envelope expression plasmid (GeneCopoeia), and pHAGE-CMV-Luc2-IRES-ZsGreen-W transfer vector plasmid (a kind gift from Jesse Bloom) were transfected into each plate at 42.19 $\mu$g, 60.94 $\mu$g, and 60.94 $\mu$g, respectively. Transfections were performed using the CalPhos mammalian transfection kit (TaKaRa Bio), and plates were incubated in a 5% $CO_2$ incubator at 33°C for 16 h with DMEM10. Following incubation, the medium was replaced with 11 mL of fresh DMEM10 and incubated for an additional 18 to 24 h.

The culture supernatants were clarified by centrifugation at 500 $\times$ $g$ and 4°C for 5 min using a Sorvall ST-40R centrifuge and TX-1000 rotor. The supernatants were pooled and filtered using a 0.45-$\mu$m polyethersulfone (PES) filter (Thermo Fisher Scientific, Waltham, MA) and ultracentrifuged in ultraclear round-bottom tubes (Fisher Scientific) using an Optima L-90K ultracentrifuge and a SW 32 Ti swinging-bucket rotor at 16°C for 2.5 h at 50,000 $\times$ $g$. The pellets were resuspended in 1$\times$ phosphate-buffered saline (PBS), aliquoted, and stored at $-$80°C.

**SARS-CoV-2 lentivirus surrogate neutralization assay.** Neutralization was measured by the reduction of luciferase expression for samples incubated with pseudotyped lentivirus relative to luciferase expression in control wells containing only SARS-CoV-2-pseudotyped lentivirus and cells. The half-maximal inhibitory concentration ($IC_{50}$) was used as the reportable value for the SCLSNA and was generated using GraphPad Prism v.9.3 software. Sample dilutions were logarithm transformed ($log_{10}$), and all raw data were normalized to a common scale (20). For data normalization, wells containing pseudotyped lentivirus plus HEK293T/ACE2-TMPRSS2 cells were defined as 0% neutralization, and wells containing only HEK293T/ACE2-TMPRSS2 cells were defined as 100% neutralization. A nonlinear regression curve was used to determine the $IC_{50}$ values for the samples once the relative luminescence units (RLU) decreased to half the response of the virus control wells.

In preparation for the SCLSNA, HEK293T/ACE2-TMPRSS2 cells were seeded at 1 $\times$ $10^4$ cells/mL in poly-L-lysine-precoated plates (Corning, Glendale, AZ). Cells were incubated in a 5% $CO_2$ incubator at 37°C for 18 to 24 h prior to performing the assay. The test samples were diluted 1:20, followed by an eight-step 2-fold serial dilution. After the addition of pseudotyped lentiviruses, plates containing serially diluted test sample and pseudovirus were incubated for 1 h at 37°C. Prior to HEK293T/ACE2-TMPRSS2 cell infection, the DMEM10 cell culture medium was replaced with DMEM containing 5% FBS and 5 $\mu$g/mL Polybrene. The diluted test sample containing pseudovirus mixture was transferred to the cell plate and incubated under 5% $CO_2$ at 37°C for 48 h. Luminescence was detected using the Bright-Glo luciferase assay system (Promega, Madison, WI) and a Biotek Cytation 1 imaging reader. Raw data were obtained using a Biotek Gen5 microplate reader, and data analysis was performed using GraphPad Prism v.9.3.

**SARS-CoV-2 plaque reduction neutralization test.** The SARS-CoV-2 PRNT was adapted from a previously described method for SARS-CoV-1 (41). Briefly, serially diluted serological specimens were mixed with diluted SARS-CoV-2 at 100 PFU/100 $\mu$L in a 96-well plate. The antibody-virus mixture was added in duplicate to 12-well plates containing preplated Vero E6 cells. All plates were incubated at 37°C with 5% $CO_2$ for 1 h of adsorption, followed by the addition of a liquid overlay. The liquid overlay was removed after a 3-day incubation, and the cells were fixed with 10% neutral buffered formalin. The monolayer in each well was stained with 0.5% (wt/vol) crystal violet, and the average number of plaques was counted for each dilution. The reciprocal of the highest dilution resulting in at least 50% and 90% reduction in plaques (compared with controls) were defined as the $PRNT_{50}$ and $PRNT_{90}$, respectively. $PRNT_{50}$ and $PRNT_{90}$ values of $\geq$20 were considered positive for SARS-CoV-2 neutralizing antibodies, whereas titers of <20 were considered negative (8).

**Statistical analysis and visualization.** Neutralization was determined by $IC_{50}$ once the plasma samples reduced the RLU by 50% relative to the virus control wells. Plasma sample dilutions were log-transformed, normalized, and plotted using nonlinear regression to obtain the $IC_{50}$ values. Based on the FDA guidelines, the sample suitability acceptance criteria were set at a 20% coefficient of variation (CV) between sample replicates and a goodness of fit ($R^2$) of $\geq$0.700 (22). The assay suitability acceptance criteria within the virus and cell control replicates for each assay were set at 30% CV and a difference of $\geq$1,000$\times$ (at least 3 $log_{10}$ above the background) between the virus control and cell control RLU (19). All data analysis was performed using GraphPad Prism v.9.3 software.

The clinical specificity and sensitivity of the SCLSNA were compared to those of the gold-standard PRNT assay. Sixty samples from the NML COVID-19 National Panel that tested positive for SARS-CoV-2 neutralizing antibodies (NAbs) and 60 pre-COVID-19 samples negative for SARS-CoV-2 NAbs were used in the comparison. Contingency tables were generated to calculate the sensitivity and specificity values.

Repeatability (intra-assay precision) was examined to measure the degree of agreement between results from different assays of the same homogenous sample material (23). Three different concentrations were used and classified as high ($PRNT_{50}$, 1:$\geq$640), mid ($PRNT_{50}$, 1:160), and low ($PRNT_{50}$, 1:40) titers, based on our in-house $PRNT_{50}$ titer results. Each sample was processed in triplicate during two separate weeks for a total of six determinations each. The analysts performed the assay using the same equipment and test conditions each week within approximately the same time frame.

Reproducibility (interassay variability) was examined to measure the degree of agreement between

individual results using the same homogenous sample material from different analysts. Three different concentrations were assessed on different days between different analysts.

Robustness was determined by examining the ability of the SCLSNA to provide analytical results of acceptable accuracy and precision under different conditions. Three concentrations of test samples were tested in triplicate during two different weeks for a total of six determinations for each sample. The different operational conditions tested included a comparison between the Agilent BioTek Cytation 1 imaging reader and Promega's GloMax Navigator.

The assay performance and acceptance criteria were based off the percent CV, which measures relative variability. The acceptance range used throughout the validation for the repeatability, reproducibility, and robustness was ≤20%, following FDA guidelines (22).

To determine linearity within the SCLSNA, a WHO international reference panel for anti-SARS-CoV-2 immunoglobulin was used to assess the ability of the assay to produce results that are directly proportional to the concentration of an analyte. The reference panel (NIBSC) consisted of pooled plasma from individuals from the United Kingdom or Norway who had recovered from COVID-19 (42). The negative control consisted of pre-COVID-19 plasma from healthy blood donors, collected before 2019.

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
