## [Reviewer comments · Microbiology Spectrum]

Microbiology Spectrum

Validation and Establishment of a SARS-CoV-2 Lentivirus Surrogate Neutralization Assay as a Pre-Screening tool for the Plaque Reduction Neutralization Test

John Merluza, Johnny Ung, Kai Makowski, Alyssia Robinson, Kathy Manguiat, Nicole Mueller, Jonathan Audet, Julie Chih-yu Chen, James Strong, Heidi Wood, and Alexander Bello

Corresponding Author(s): Alexander Bello, National Microbiology Laboratory

Review Timeline:

Submission Date:	September 16, 2022
Editorial Decision:	October 14, 2022
Revision Received:	December 6, 2022
Accepted:	December 7, 2022

Editor: Juan Ludert

Reviewer(s): Disclosure of reviewer identity is with reference to reviewer comments included in decision letter(s). The following individuals involved in review of your submission have agreed to reveal their identity: Anton M Sholukh (Reviewer #1)

Transaction Report:

DOI: <https://doi.org/10.1128/spectrum.03789-22>

October 14, 2022

Dr. Alexander Bello
National Microbiology Laboratory
Special Pathogens
Winnipeg, Manitoba
Canada

Re: Spectrum03789-22 (Validation and Establishment of a SARS-CoV-2 Lentivirus Surrogate Neutralization Assay as a Pre-Screening tool for the Plaque Reduction Neutralization Test)

Dear Dr. Alexander Bello:

Thank you for submitting your manuscript to Microbiology Spectrum. The manuscript was reviewed by two experts who agree that it is of interest to the field. However, they also raised some concerns aimed to improve your manuscript. When submitting the revised version of your paper, please provide (1) point-by-point responses to the issues raised by the reviewers as file type "Response to Reviewers," not in your cover letter, and (2) a PDF file that indicates the changes from the original submission (by highlighting or underlining the changes) as file type "Marked Up Manuscript - For Review Only". Please use this link to submit your revised manuscript - we strongly recommend that you submit your paper within the next 60 days or reach out to me. Detailed instructions on submitting your revised paper are below.

Link Not Available

Sincerely,

Juan E. Ludert

Journals Department
Reviewer comments:

Reviewer #1 (Comments for the Author):

Manuscript by Merluza et al., Validation and Establishment of a SARS-CoV-2 Lentivirus Surrogate Neutralization Assay as a Pre-Screening tool for the Plaque Reduction Neutralization Test

The study is important to the field as it addresses qualification of the pseudovirus-based neutralization assay for SARS-CoV-2 in comparison to the PRNT, which is considered gold standard but not accessible in every lab due to safety concerns. Pseudovirus-based assay is currently used widely in different laboratories and facilities in natural history studies and vaccine trials while PRNT was the major method during early COVID-19 pandemic. As pseudovirus-based technologies are replacing PRNT in many studies and especially clinical labs, qualification and validation of pseudovirus assay is important to maintain

computability and convertibility of the results between different labs and studies.

Major points:

1. Title of the presented manuscript states that pseudovirus neutralization assay is considered as a "pre-screening tool" for PRNT. No significant reasoning in Introduction or throughout the manuscript is provided why PRNT would require sample prescreening rather than to reduce workload. Moreover, it has been demonstrated in multiple studies that results of both assays are highly correlative and therefore pseudovirus-based assays are widely used in the field to replace PRNT. Suggestion: either reasoning and data (see below) should be provided to justify SCLSNA as a pre-screening tool or the title of the article should be adjusted to reflect provided data.

2. It is not clear how positivity threshold for SCLSNA was established. If that was a % neutralization what was the exact value of it? As SCLSNA is assessed in comparison to PRNT for which positivity threshold was 20% neutralization these data are important to understand why SCLSNA was less sensitive for B.1.1.7 variant compared to PRNT.

3. It is not clear why two luminometers that were used in the study to assess robustness are referred as few or several, this is misleading as well as a statement on lines 273-274 reading as "under variety of conditions". What was that variety exactly? Which conditions were tested using two luminometers? As 2 luminometers were set to test, what is the difference between luminometers? Subsequent discussion of these data also exaggerates the number of devices and conditions used (lines 332-340).

As robustness of an assay is intended to test capability of the assay to yield similar outcomes under different assay conditions, a reading device is often the least influential component. As devices measuring the same parameter, e.g., luminescence, absorbance, fluorescence, etc., made by different manufacturers still use the same physical principle to operate, sample composition affects the final readout at much greater extent. To account for that lipemic, icteric or hemolyzed serum/plasma samples are used to analyze assay robustness. If that was not tested and it is impossible to conduct additional measurements, it should be discussed as one of the study limitations.

4. Why specifically B.1.1.7 variant was used? This strain is not actively circulating, it is not considered as neutralization escape variant and as pointed out in Shen et al., (ref. 35 in the manuscript), fold decline between Wuhan and B.1.1.7 for convalescent sera was 1.5-fold for ID50 which is much less than it was for Beta, Delta and Omicron variants. Discussion states that "To determine correlation between the SCLSNA and PRNT using SARS-CoV-2 convalescent patient samples, we conducted a correlation assessment, but the correlation coefficient was low (data not shown)." Does this, taken together with 88% sensitivity imply that neutralization was detected in PRNT but in SCLSNA? If so, data from PRNT in comparison to SCLSNA should be provided as table or graph. It is unclear why PRNT which known as less sensitive assay detected neutralization against B.1.1.7 variant better than pseudovirus assay.

5. Discussion is unreasonably long, 3 pages (107 lines)! There are several iterative paragraphs describing subjectivity, laborious nature and BSL3 requirements for PRNT but no reasoning why SCLSNA is used as a pre-screening tool for PRNT have been provided. Is it to exclude high positive samples from PRNT thus reducing workload?

Why authors discuss end-point titer in relation to PRNT as it measures same IC50 as pseudovirus-based assay?

Minor points:

1. Line 137: what type of nonlinear regression was used? If it is different for different analyses, then it should be specified in the appropriate sections.

2. Line 197-198: What are the "different operational conditions" for the luminometers used?

3. Line 217; It seems that text should read "Figure 1A". Figure panels are often not specified in the text.

4. Table 1. Brining IC50 values to the format e.g. $N \times 10^n$ will help processing the data.

5. Based on the data from Table 1, the best seeding condition was 1.3×10^4 , why then 10^4 was selected for future experiments? Was it tested the same way as other cell seeding densities?

6. Line 235: Should read Figure 1B?

7. Reference #29 is incomplete.

Reviewer #2 (Comments for the Author):

The Covid pandemic has spearheaded the development and improvement of diagnostic and analytic tools. This is an example of this trend. In the present paper, the authors propose a SCLNA neutralization technique that can be used in biosafety Level 2 laboratories, and does not require special training, since it replaces SARS-Cov-2 virus with pseudotyped S protein-expressing lentivirus. The relevant contributions are that the results are comparable with the standard PRNT technique and reproducible. It also complies with previous basic parameters.

A point that needs to be addressed is that the authors mention a limitation using this technique in other laboratories to assess reproducibility, and instead used different instruments to measure luminescence. Since they are selling it as an easy and flexible

method this is an important aspect to assure accuracy and reproducibility.

Although not original, in general, I consider it is a worthwhile work that will provide yet another method to diagnose Covid-19, and assist in the health decision making regarding, among other matters, the suitability of vaccines.

Staff Comments:

Preparing Revision Guidelines

Please return the manuscript within 60 days; if you cannot complete the modification within this time period, please contact me. If you do not wish to modify the manuscript and prefer to submit it to another journal, please notify me of your decision immediately so that the manuscript may be formally withdrawn from consideration by Microbiology Spectrum.

Reviewer #1

1. Title of the presented manuscript states that pseudovirus neutralization assay is considered as a "pre-screening tool" for PRNT. No significant reasoning in Introduction or throughout the manuscript is provided why PRNT would require sample prescreening rather than to reduce workload. Moreover, it has been demonstrated in multiple studies that results of both assays are highly correlative and therefore pseudovirus-based assays are widely used in the field to replace PRNT. Suggestion: either reasoning and data (see below) should be provided to justify SCLSNA as a pre-screening tool or the title of the article should be adjusted to reflect provided data.

Thank you for your comments and suggestions on the title of our manuscript. We agree that the title needs to be revised to more directly address the content in our data. The initial intent of the SCLSNA was to design the method as a pre-screening approach to identify negative and low neutralizing samples where once identified, can be excluded for PRNT testing and reduce the overall workload. Over time, the purpose of the SCLSNA has evolved to support and compare results from our in-house PRNT testing with the SCLSNA and in some studies, the SCLSNA has been used as the primary measure for neutralization analysis. As a result, we will change the title in our manuscript from "Validation and Establishment of a SARS-CoV-2 Lentivirus Surrogate Neutralization Assay as a Pre-Screening tool for the Plaque Reduction Neutralization Test" to "Validation and Establishment of a SARS-CoV-2 Lentivirus Surrogate Neutralization Assay as an Alternative Approach to the Plaque Reduction Neutralization Test" in order to better reflect the overall purpose of the manuscript.

2. It is not clear how positivity threshold for SCLSNA was established. If that was a % neutralization what was the exact value of it? As SCLSNA is assessed in comparison to PRNT for which positivity threshold was 20% neutralization these data are important to understand why SCLSNA was less sensitive for B.1.1.7 variant compared to PRNT.

Thank you for your comment on the positivity threshold. During the validation, we opted to calculate the LOD and LOQ of the SCLSNA to determine the cut off values for negative samples that closely matched the negative results of the PRNT₅₀. Based on this assessment, we assigned any values that were below an IC₅₀ of 20 as a negative sample with a nominal value of 10 and any values above an IC₅₀ of 20 were reported as a positive sample. Although not a direct determination of the positivity threshold, we based our positive samples from the LOD instead. We have included this explanation in the results section (Lines 309 to 315):

"The standard deviation determined from the mean IC₅₀ values of the negative samples resulted in a LOD of 19.60 and a LOQ of 65.32 which were three and ten times the standard deviation respectively. We used a cut-off of < 20 for negative samples and assigned them a nominal value of 10. This was done to distinguish negative from positive results in our qualitative representation of our results. An upper limit of quantification (ULOQ) was not determined during the validation as we opted to calculate the LOD and LOQ of the SCLSNA instead."

3. It is not clear why two luminometers that were used in the study to assess robustness are referred as few or several, this is misleading as well as a statement on lines 273-274 reading as "under variety

of conditions". What was that variety exactly? Which conditions were tested using two luminometers? As 2 luminometers were set to test, what is the difference between luminometers? Subsequent discussion of these data also exaggerates the number of devices and conditions used (lines 332-340).

As robustness of an assay is intended to test capability of the assay to yield similar outcomes under different assay conditions, a reading device is often the least influential component. As devices measuring the same parameter, e.g., luminescence, absorbance, fluorescence, etc., made by different manufacturers still use the same physical principle to operate, sample composition affects the final readout at much greater extent. To account for that lipemic, icteric or hemolyzed serum/plasma samples are used to analyze assay robustness. If that was not tested and it is impossible to conduct additional measurements, it should be discussed as one of the study limitations.

Thank you for your comments regarding robustness and the use of different luminometers. The luminometers tested were used to evaluate operational conditions as a measure of robustness and the variety refers to the two different luminometers that were tested. All devices used were configured to the same settings to focus on the differences in RLU intensity. We mentioned the difference between the devices in lines 344 to 346 and 348 to 351 of the manuscript:

“Another strategy we employed in our validation was the use of two different luminometers to achieve robustness; we were able to achieve acceptable robustness from the low variation seen across different luminometers.”

“The Promega GloMax® Navigator is used solely for the detection of bioluminescence whereas the Agilent BioTek Cytation 1 is a multimode reader that combines fluorescence and high contrast brightfield imaging.”

We agree that a validation study typically does not use the reading device as a parameter for evaluation and will likely have the least amount of influence in our results. Further explanation is found in lines 355 to 361 of the Discussion:

“In our preliminary studies, we have noticed differences in RLU signals between different detection devices such as the Agilent BioTek Cytation 1, GloMax® Navigator and the Agilent BioTek Synergy (data not shown). Based on these differences, we wanted to demonstrate that the SCLSNA can still achieve similar IC_{50} values between different reading devices and confirm that using different devices had no impact in obtaining the IC_{50} . For these reasons we felt it was important to evaluate the operational conditions as our main parameter in evaluating robustness.”

Alternatively, we have performed the SCLSNA with different analysts on different days and in different laboratories which are test conditions normally tested in a robustness evaluation. These results indicate comparable data between test conditions achieving below 20% CV. However the samples tested were not from the same panel (NML CNP) used in the validation study because we had a limited supply of these NML CNP samples. This approach might typically be seen in a validation but given the discrepancies we were previously seeing with detection devices we felt that a robustness evaluation

based on the detection device was necessary in this context. We have included these explanations in the Discussion section of the manuscript from lines 352 to 361:

“One limitation in our robustness analysis was the inability to compare assay performance in different laboratories due to logistical challenges and unavailability at the time of the study. We opted to test robustness using different luminescence detectors as an alternative and to confirm the reliability of the assay. In our preliminary studies, we observed differences in RLU signals between different detection devices such as the Agilent BioTek Cytation 1, GloMax® Navigator and the Agilent BioTek Synergy (data not shown). Based on these differences, we wanted to show that the SCLSNA can still achieve similar IC_{50} values between different reading devices and confirm that using different devices had no impact in obtaining the IC_{50} . For these reasons we felt it was important to evaluate the operational conditions as our main parameter in evaluating robustness.”

4.

1) Why specifically B.1.1.7 variant was used? This strain is not actively circulating, it is not considered as neutralization escape variant and as pointed out in Shen et al., (ref. 35 in the manuscript), fold decline between Wuhan and B.1.1.7 for convalescent sera was 1.5-fold for ID50 which is much less than it was for Beta, Delta and Omicron variants.

At the time the study was conducted, the B.1.1.7 was the dominant strain and this variant was used to test samples from the National Microbiology Laboratory NML COVID-19 National Panel (NML CNP). Unfortunately the NML CNP panel contained a limited supply and we were only able to test against the B.1.1.7 variant and not the other VOCs as you mentioned afterwards. We included this explanation in lines 389 to 394 of the Discussion session in the manuscript:

“The NML COVID-19 National Panel sample set used in this study consisted mainly of one antibody titer range at lower neutralization titers (1:80 PRNT₅₀) from convalescent donors who were naturally infected with SARS-CoV-2 before the vaccine was available, making it difficult to achieve a proper correlation analysis. As a result, we used clinical sensitivity and specificity as a measure for comparison and were able to show good comparability to the PRNT, as seen in other studies...”

2) Discussion states that "To determine correlation between the SCLSNA and PRNT using SARS-CoV-2 convalescent patient samples, we conducted a correlation assessment, but the correlation coefficient was low (data not shown)." Does this, taken together with 88% sensitivity imply that neutralization was detected in PRNT but in SCLSNA? If so, data from PRNT in comparison to SCLSNA should be provided as table or graph. It is unclear why PRNT which known as less sensitive assay detected neutralization against B.1.1.7 variant better than pseudovirus assay.

The NML CNP panel consisted of samples collected early in the COVID-19 pandemic (from May 13, 2020 to August 22, 2020) containing samples from patients were not yet administered the COVID-19 vaccine, resulting in a majority of the samples containing lower titres (1:80) as determined through PRNT. When the NML CNP was tested through the SCLSNA, neutralization was also detected but more precise IC_{50} values were generated that were either slightly higher or lower than the reported values of the PRNT.

The differences in titres resulted in a low correlation coefficient as it was difficult to compare to the PRNT, which gave a low value of 1:80 for a majority of the samples. However, when we evaluated the two platforms based on the presence or absence of neutralization, a clinical sensitivity and specificity was a more suitable approach in assessing neutralization.

5.

1) Discussion is unreasonably long, 3 pages (107 lines)! There are several iterative paragraphs describing subjectivity, laborious nature and BSL3 requirements for PRNT but no reasoning why SCLSNA is used as a pre-screening tool for PRNT have been provided. Is it to exclude high positive samples from PRNT thus reducing workload?

Thank you for your comments regarding the discussion. We excluded some sections of the discussion and condensed it as much as we could. However, our discussion included more content to ensure we cover all aspects of the validation justifying our approach to the validation parameters along with the direct comparison with the PRNT. We felt if we were to exclude some areas then this would result in the lack of clarity for our readers in some sections.

Similar to our response in question 1, initially the SCLSNA was used to reduce the workload of the PRNT testing by verifying negative samples from the PRNT with the SCLSNA. Depending on the study, we now use the SCLSNA in parallel to the PRNT or test a smaller subset of samples to confirm the results from the PRNT. We have changed the title of this manuscript to reflect that the SCLSNA is an alternative approach rather than a pre-screening method to the PRNT.

2) Why authors discuss end-point titer in relation to PRNT as it measures same IC50 as pseudovirus-based assay?

Thank you for this question regarding the end-point titer. We made reference to end-point titer as one of the advantages of the SCLSNA over the PRNT. With the PRNT, we assign a cut off PRNT₅₀ dilution of 1:640 for all high positive samples and the exact PRNT₅₀ for these samples were not determined. The 8-step dilution series in the SCLSNA allows us to take any given sample to an end-point dilution thus enabling us to determine a more accurate IC₅₀ value.

Minor points:

1. Line 137: what type of nonlinear regression was used? If it is different for different analyses, then it should be specified in the appropriate sections.

This type of nonlinear regression is based off a dose response curve where we took the logarithm of the X values in this case these are the serial dilutions of the samples tested and normalized the Y values or RLU signals to determine our IC₅₀ values. The same type of nonlinear regression was used for all samples tested. We included this explanation in lines 138 to 142 in the Materials and Methods in the manuscript:

“This type of nonlinear regression is based off of a dose response curve where we took the logarithm of the X values in this case the serial dilutions of the samples tested and normalized the Y values or RLU signals to determine our IC₅₀ values. The same type of nonlinear regression was used for all samples tested.”

2. Line 197-198: What are the "different operational conditions" for the luminometers used?

The different conditions tested were in reference to the operational condition. In this case, the different operational conditions were the use of two different luminometers - the Agilent BioTek Cytation1 and Promega's Glo-Max® Navigator. The revised statement is as follows (Line 199-200): “The different operational conditions tested include the comparison between the Agilent BioTek Cytation1 and Promega's Glo-Max® Navigator.”

3. Line 217; It seems that text should read "Figure 1A". Figure panels are often not specified in the text.

Thank you for your correction. The text has been revised in line 219.

“A high titer sample ($\geq 1:640$ PRNT₅₀) was tested against an ancestral Wuhan spike pseudotyped lentivirus with nine cell seeding densities ranging from 7.8×10^2 to 2.0×10^5 cells/well (Figure 1A).”

4. Table 1. Bringing IC50 values to the format e.g. N x 10^n will help processing the data.

Thank you for the correction. We have revised Table 1 to the “Nx10^n” format.

5. Based on the data from Table 1, the best seeding condition was 1.3×10^4 , why then 10^4 was selected for future experiments? Was it tested the same way as other cell seeding densities?

Yes, we agree that based on the R₂ value the 1.3×10^4 is better than the 1.0×10^4 seeding density used in our experiments; however we were targeting optimal RLU signal and IC₅₀ values and the 1.0×10^4 seeding density gave us the best and most consistent results. The reasoning for our approach can be found in lines 219 to 226 in the Results section of the manuscript:

“The selection for the optimal cell density was based on the combination of the cell density ($> 1 \times 10^3$ cells/well), IC₅₀ (> 640) and goodness of fit ($R^2 > 0.9$). The results indicate IC₅₀ values > 640 and $R^2 > 0.9$ for cell densities between 7.8×10^2 to 6.3×10^3 , but these seeding densities were not selected due to the potential of increased variability in SCLSNA testing observed in the lower cell densities (9). The cell densities above 1×10^4 cells/well demonstrated fit our criteria for cell density, IC₅₀ and goodness of fit. Thus, the cell density of 1.0×10^4 cells/well was selected as an optimal cell seeding density for the SCLSNA (Table 1) and this seeding density was used throughout the study.”

6. Line 235: Should read Figure 1B?

Previously line 235 is now line 248-249 is in reference to the samples tested in the clinical specificity and sensitivity assessment found in Table 2. Lines 237 to 241 are in reference to Figure 1B.

7. Reference #29 is incomplete.

Revision complete.

Reviewer #2 (Comments for the Author):

The Covid pandemic has spearheaded the development and improvement of diagnostic and analytic tools. This is an example of this trend. In the present paper, the authors propose a SCLNA neutralization technique that can be used in biosafety Level 2 laboratories, and does not require special training, since it replaces SARS-Cov-2 virus with pseudotyped S protein-expressing lentivirus. The relevant contributions are that the results are comparable with the standard PRNT technique and reproducible. It also complies with previous basic parameters.

Thank you for highlighting the goal of our manuscript. The approach of validating the method and performing a direct comparison with the PRNT along with the previous work from other studies further confirms the strength of the SCLNA. Moving forward, our goal within our laboratory and amongst our collaborators within the National Microbiology Laboratory is to establish the SCLNA as an alternative approach to the PRNT that can produce reliable results with a faster turnaround time.

A point that needs to be addressed is that the authors mention a limitation using this technique in other laboratories to assess reproducibility, and instead used different instruments to measure luminescence. Since they are selling it as an easy and flexible method this is an important aspect to assure accuracy and reproducibility.

This is a good point regarding the reproducibility and an area that we continue to focus on as the SCLNA method is being transferred across different laboratories. As mentioned with Reviewer #1, there were differences in RLU shown across different luminescence used which caused us to examine this area further by including it as an approach to the validation study. A comparative analysis between reading devices is not typically seen in evaluating robustness but given the discrepancies seen across different devices we felt that this was necessary. The outcome of this robustness evaluation showed no significant differences between devices but we are also aware that such differences exist and need to be accounted for in future studies.

Although not original, in general, I consider it is a worthwhile work that will provide yet another method to diagnose Covid-19, and assist in the health decision making regarding, among other matters, the suitability of vaccines.

December 7, 2022

Dr. Alexander Bello
National Microbiology Laboratory
Special Pathogens
Winnipeg, Manitoba
Canada

Re: Spectrum03789-22R1 (Validation and Establishment of a SARS-CoV-2 Lentivirus Surrogate Neutralization Assay as a Pre-Screening tool for the Plaque Reduction Neutralization Test)

Dear Dr. Alexander Bello:

I am pleased to let you know that your manuscript has been accepted, and I am forwarding it to the ASM Journals Department for publication. You will be notified when your proofs are ready to be viewed.

Sincerely,

Juan E. Ludert
Editor, Microbiology Spectrum
